# Effectiveness of an Active and Continuous Surveillance Program for Intensive Care Units Infections Based on the EPIC III (Extended Prevalence of Infection in Intensive Care) Approach

**DOI:** 10.3390/jcm11092482

**Published:** 2022-04-28

**Authors:** Giorgia Montrucchio, Gabriele Sales, Giulia Catozzi, Stefano Bosso, Martina Scanu, Titty Vita Vignola, Andrea Costamagna, Silvia Corcione, Rosario Urbino, Claudia Filippini, Francesco Giuseppe De Rosa, Luca Brazzi

**Affiliations:** 1Department of Surgical Sciences, University of Turin, 10126 Turin, Italy; gabriele.sales@unito.it (G.S.); andrea.costamagna@unito.it (A.C.); claudia.filippini@unito.it (C.F.); luca.brazzi@unito.it (L.B.); 2Department of Anaesthesia, Critical Care and Emergency, Città Della Salute e Della Scienza Hospital, Corso Dogliotti 14, 10126 Turin, Italy; martiscanu89@gmail.com (M.S.); rurbinocsst@gmail.com (R.U.); 3Department of Health Sciences, University of Milan, 20122 Milan, Italy; giuliacatozzi.ds@gmail.com; 4Department of Anesthesiology and Critical Care, “Cardinal Massaia” Hospital, 14100 Asti, Italy; steo_bos@libero.it; 5Anesthesia and Intensive Care Unit, San Giovanni Bosco Hospital, 10154 Turin, Italy; tittyvitavignola@outlook.it; 6Department of Medical Sciences, Infectious Diseases, University of Turin, 10126 Turin, Italy; silvia.corcione@unito.it (S.C.); francescogiuseppe.derosa@unito.it (F.G.D.R.); 7School of Medicine, Tufts University, Boston, MA 02111, USA

**Keywords:** infections, intensive care unit, antimicrobial stewardship, infection control, drug resistance, bacterial, point prevalence study

## Abstract

We evaluated the effectiveness of the Extended Prevalence of Infection in Intensive Care (EPIC) III data collection protocol as an active surveillance tool in the eight Intensive Care Units (ICUs) of the Intensive and Critical Care Department of the University Hospital of Turin. A total of 435 patients were included in a six-day study over 72 ICU beds. 42% had at least one infection: 69% at one site, 26% at two sites and 5% at three or more sites. ICU-acquired infections were the most common (64%), followed by hospital-associated infections (22%) and community-acquired (20%), considering that each patient may have developed more than one infection type. 72% of patients were receiving at least one antibiotic: 48% for prophylaxis and 52% for treatment. Mortality, the length of ICU and hospital stays were 13%, 14 and 29 days, respectively, being all estimated to be significantly different in patients without and with infection (8% vs. 20%; 4 vs. 20 and 11 vs. 50 (*p* < 0.001). Our data confirm a high prevalence of infections, sepsis and the use of antimicrobials. The repeated punctual prevalence survey seems an effective method to carry out the surveillance of infections and the use of antimicrobials in the ICU. The use of the European Centre for Disease Prevention and Control (ECDC) definitions and the EPIC III protocol seems strategic to allow comparisons with national and international contexts.

## 1. Introduction

Infections are a major cause of admissions and prolonged stays in intensive care units (ICUs). They affect approximately 30% of patients, with large variations between different geographical regions [1,2,3,4,5,6,7], and they are the leading cause of death in non-cardiac ICUs, with still very high mortality rates and associated costs [8,9].

Sepsis and septic shock can complicate both community-acquired infections, which account for up to 70% of all cases of sepsis [8], and healthcare-associated infections (HAI), which would be mostly preventable by adequate infection prevention and control (IPC) measures [10,11,12].

Although extremely variable in the literature, data regarding the real prevalence of HAIs remains high in Europe (6.5%) [13,14], with values probably much higher in ICUs. Unfortunately, many articles do not report the differentiation between community and hospital-acquired sepsis, leading to a possible underestimation of the impact of HAIs, however potentially prevented in about 55% of cases by the implementation of multifaceted IPC interventions [15,16,17].

Epidemiological information on the underlying source of infections, associated microorganisms, treatment and outcomes are essential to identify gaps and optimize patient management. Unfortunately, although surveillance systems have been proposed at local [18] and international levels [19,20], adherence to them is not uniform in terms of both data collection and definitions [21], and this limits the comparability of the data over time. In particular, the integration between infection and/or colonization systematic data collection, control measures, and their application and evolution over time is complex. Moreover, data complexity does not allow their timely use, given the long processing and interpretation times, partially limiting the possibility of continuous and proactive surveillance. Another point to be considered is the lack of local comparisons, on a national or regional basis, capable of reflecting the specific characteristics of the population, the intensity of care, as well as the microbiological trend of the local ecology.

In this scenario, the use of punctual prevalence studies, which are more easily achievable and repeatable over time, has been proposed, especially in ICUs. Their validity and reliability, however, might be limited, given the method and timing of the data collection used [22].

Recently, a worldwide study [9] collected comprehensive data on the global epidemiology of ICU infections in 1150 centers in 88 countries, reporting that 54% of admitted patients had suspected or proven infection, 70% received at least one antibiotic, and Gram-negative bacteria were the predominant microorganisms (67%). One of the strengths of this study was the use of an exhaustive but essential data collection protocol, widely applicable in different contexts, which guaranteed great participation and reliability of the collected data.

As valid epidemiological data are needed to increase the awareness of the impact of infection among ICU patients, we applied the EPIC III protocol to estimate the prevalence of community and hospital-associated infections, associated risk factors and distribution of antimicrobial use in the ICUs of the Intensive and Critical Care Department of the University Hospital of Turin. We also evaluated the effectiveness of this data collection protocol as an active surveillance tool.

## 2. Materials and Methods

### 2.1. Study Design

This is a 24-h prospective observational point prevalence study, with repeated observations every 2 months. Surveillance was carried out in all medical/surgical ICUs of the Department of Anesthesia and Resuscitation of the Città della Salute e della Scienza Hospital of Turin for a total of 8 ICUs and 72 ICU beds.

The study was approved by the local ethics committee (prot. No.0000255), and informed consent was obtained from each patient enrolled.

The overall duration of the study was 1 year; each observation lasted 24 h, and the follow-up for the outcome was performed at 60 days, regardless of the patient location. Six observations were performed throughout the year, evenly distributed over 12 months. Data were recorded for all patients present or admitted to ICU during the 24-h periods of study, from 1 December 2017, 08:00 to 2 December 2018, 07:59.

All patients hospitalized or admitted to ICU on one of the days of the study were involved, with no exclusion criteria, except for the absence of informed consent.

### 2.2. Study Context

All ICUs were able to perform blood cultures or qualitative respiratory cultures. Intermittent and continuous renal replacement therapies, high nasal oxygen flow, echocardiography and invasive monitoring were available in all units and extracorporeal membrane oxygenation (ECMO) in two units. An infectious disease specialist or clinical microbiologist was available 12 h a day, 5 days a week, and on-call during nights and weekends. Therapeutic drug monitoring was available for vancomycin, voriconazole, aminoglycosides and beta-lactams.

### 2.3. Data Collection

Data was collected using the case report form (CRF; see Appendix A) used in EPIC III, investigating presence of infection (up to a maximum of four per patient).

### 2.4. Operative Definitions

European Centre for Disease Prevention and Control (ECDC) case definitions were applied for infection surveillance [21]. Sepsis and septic shock were defined according to the Third International Consensus Definitions [23]. Multi-drug resistant organisms were defined according to ECDC 2012 definitions [24].

In case of infection, clinicians were asked to classify the mode of acquisition as certainly/possibly/probably and community-acquired/hospital-acquired/ICU-acquired [9].

Infections occurring at least 48 h after hospital admission were defined as ‘hospital-acquired’. Infections occurring at least 24 h after ICU admission were defined as ‘ICU acquired’. All other infections were defined as ‘community-acquired’.

Antimicrobial prophylaxis (not previously defined in the EPIC protocol) was clinically defined as the use of an antimicrobial to prevent the occurrence of an infection, both in medical or surgical contexts.

### 2.5. Outcomes

Primary outcomes were hospital and 60-days all causes of death. Secondary outcomes were ICU and hospital length of stay (LoS).

### 2.6. Statistical Analysis

Continuous data are reported as mean and standard deviation (SD) or median and interquartile range (IQR) as appropriate; categorical data are reported as number and percentage. For continuous variables, a comparison between two groups was performed using the unpaired student’s t-test or Wilcoxon–Mann–Whitney test depending on type of distribution; for categorical variables, Chi-square test or Fisher’s exact test was used as appropriate. Comparison of continuous variables between more than two groups was conducted using Kruskal–Wallis test.

A multivariable logistic regression model was performed using infection as dependent variable and choosing the following covariates resulting significant in the univariate analysis: reason for admission, cardiovascular disease, sex, age, invasive ventilation, vasopressors, central venous access, dialysis and Chronic Obstructive Pulmonary Disease (COPD).

To evaluate possible risk factors for death (60 days mortality), demographic and clinical characteristics associated with mortality were selected as covariates to compete in a multivariable logistic regression model with backward selection.

Results were expressed by calculating the Odd Ratio (OR) and a 95% confidence interval.

All statistical tests were two-sided. *p* values of 0.05 or less were considered statistically significant and were conducted using the SAS ver. 9.4 (SAS Institute, Cary, NC, USA) and SPSS ver. 26.0 (IBM Corp., Armonk, NY, USA).

## 3. Results

A total of 435 patients were included in the six study days: 405 adults (mean age 61 years, Standard Deviation (SD) 15, range 18–87) and 30 pediatric patients (mean age 4 years, Standard Deviation (SD) 5, range 0–17). Demographic and general patient data are summarized in Table 1. Informed consent was not collected in less than 5% of patients.

Overall, 217 patients (50%) were on mechanical ventilation, 69 (16%) were in septic shock, 39 (9%) were treated with extracorporeal renal replacement and 8 (1.8%) with Extracorporeal Membrane Oxygenation (ECMO), and 114 patients (26%), received vasopressor drugs.

Characteristics of the patients according to the ICU type are shown in the Appendix A. Outcome data on mortality were present for 403 patients. The overall infected patients, according to clinical definition, were 184, whilst the total of patients with at least one positive isolate was 114.

### 3.1. Prevalence of Infections

The infection section of the CRF was completed for 425 patients (98%). A total of 184 patients (42%) had at least one infection on one of the study days: 126 patients (69%) at one site, 48 patients (26%) at two sites and 58 (32%) in more than two sites.

The proportion of infected patients was 42%, 43% and 40% in general, specialist and pediatric ICUs respectively (Table 1).

Among infected patients (184), 114 (62%) had at least one positive isolate at microbiological culture. ICU-acquired infections were the most common (117 patients—64%), followed by hospital or healthcare-associated infections (41 patients—22%) and community-acquired (36 patients—20%). Data regarding infection acquisition are reported in Table 2.

Infection characteristics according to mortality (403 patients, lacking mortality data of 32 patients) are shown in Table 3.

Considering patients with at least one positive microbiological culture (total = 114), Gram-positive bacteria were isolated in 34 patients (30%); Gram-negative bacteria were isolated in 98 (86%); 59 patients (52%) presented one or multiple multidrug-resistant (MDR) bacteria, as follows: gram-negative MDR in 47 patients (41%); gram-positive MDR in 17 patients (15%). Methicillin-resistant Staphylococcus aureus (MRSA) was isolated in 10 patients (8.8%). No cases of C. difficile (CD) have been reported. Other isolates were fungi (19 patients [17%]), viruses (8 [7%]) and anaerobes (1 [1%]). *Klebsiella* spp. was isolated in 40 patients (35%), *Pseudomonas* spp. in 30 (26%) and *Acinetobacter* spp. in 16 (14%); the total of patients with an infection caused by Carbapenem-resistant bacteria was 36 (32%). Details on sites of infection and isolated microorganisms are shown in Figure 1.

A total of 256 infections were clinically diagnosed in 184 patients overall; these were considered definite, probable or possible in 108 (59%), 44 (24%) and 52 (28%) patients, respectively.

Considering the overall number of isolates (total = 170), MDR or resistance to carbapenems were 49% and 21%, respectively, of the total of isolates.

The multivariate analysis carried out to evaluate the impact of different factors on infections evidenced that invasive ventilation, renal replacement therapy and COPD as comorbidity prior to hospitalization are all factors independently associated with an increased risk of developing an infection (Table 4A,B).

### 3.2. Antibiotic Therapy

On the six study days, 311 patients (72%) were receiving at least one antibiotic: 149 patients (48%) for medical or surgical prophylaxis and 162 (52%) for treatment. Prophylaxis was performed with one antibiotic in 101 patients (68%) and with two or more antibiotics in 48 patients (32%). Cefazolin was the most used prophylactic antibiotic (42 patients—28%), followed by amoxicillin–clavulanate (28 patients—19%) and piperacillin–tazobactam (27 patients—18%).

Antibiotic therapy was carried out with one antibiotic in 44 cases (27%), with two antibiotics in 57 cases (35%) and with three to five antibiotics in 61 cases (37%). The most frequently used molecules were meropenem (39 patients—24%), piperacillin–tazobactam (38 patients—23%) and levofloxacin (27 patients—17%). Meropenem, piperacillin–tazobactam were the most used antibiotics in patients with hospital-acquired infection (34% and 29% respectively) and ICU-acquired infection (20% and 15% respectively). Piperacillin-tazobactam, ceftriaxone and metronidazole were the most used antibiotics in patients with community-acquired infection.

### 3.3. Clinical Outcomes

Mortality of the cohort included in the present study was 13% with a statistically significant difference between patients without and with infection (8% vs. 20%; *p* < 0.001). Median LoS in ICU and hospital was 14 (IQR 4–36) and 29 (IQR 15–54) days, respectively, and was significantly different in patients without and with infection: 4 (1–12) vs. 20 (13–33) days (*p* < 0.001) and 11 (6–21) vs. 50 (22–65) days (*p* < 0.001), as shown in Table 5.

The multivariate analysis carried out to evaluate the impact of different factors on mortality at 60 days evidenced that invasive ventilation, confirmed bloodstream infection, community-acquired infection and presence of at least one comorbid condition were independently associated with a higher risk of mortality (Table 4).

## 4. Discussion

Data collected in this six-days point prevalence study, bi-monthly repeated in eight ICUs of a university hospital in Turin (Italy) between 2017 and 2018, evidenced an overall prevalence of infection of 42%. This estimate is lower than the rate found by the international EPIC III study (54%), which already showed an upward trend compared to previous EPIC studies (45% for EPIC I in 1992 [25] and 51% for EPIC II in 2007 [26]).

In our cohort, the proportion of patients with ICU-acquired infection was higher compared to the EPIC III study (26.3% vs. 21.6%). When hospital-acquired infections are also considered, we found an additional 8.9% (compared to 34.5% in the EPIC III cohort). Overall, ICU-acquired infections accounted for 64% of infections, followed by hospital-acquired (22%) and community-acquired (20%) infections, considering that each patient may have developed more than one infection type.

It is well known that HAIs represent a major patient safety issue as well as a significant economic burden, being frequently characterized by antimicrobial resistance. Among European countries, Italy is one of those where antibiotic use and prevalence of antimicrobial resistance in both the community and hospital settings are highest [14,27,28]. Even if ICU is the clinical setting in which HAI prevalence is highest, with data ranging between 19.5% in Europe [19], and 35–36.8% [29,30] in North Italy, there is a lack of specific data based on the ECDC surveillance model and repeated over time. We therefore consider the model proposed here particularly interesting for its ability to evaluate the evolution over time in the specific ecological context of reference.

Our data confirm the role that infections play in mortality. Although hospital mortality was, overall, low (12.9%) and different according to the type of ICUs (from 47.6 to 2.9%—Appendix A), the impact of infection on mortality seems notable (20.1% vs. 7.9%, *p* < 0.001).

Infections seem to obviously affect even the length of ICU and hospital stay (4 (1–12) vs. 20 (13–33) days (*p* < 0.001) and 11 (6–21) vs. 50 (22–65) days (*p* < 0.001), respectively). Mechanical ventilation, the presence of medical devices such as a central venous catheter, renal replacement therapy, ECMO and tracheostomy are all factors independently associated with an increased infections risk even if, at multivariate analysis, only invasive ventilation, renal replacement therapy and COPD were independently associated with a higher risk of infection. In line with EPIC III results, older age and the presence of at least one comorbidity were all factors independently associated with a higher risk of death in our cohort. Interestingly, multivariate analysis found that also male gender, admission from the medical department or referral from other ICUs, invasive ventilation, confirmed bloodstream infection and community as a source of infection are factors associated with an increased risk of death.

Regarding microbiological isolation, considering patients with at least one positive microbiological culture, in line with EPIC III data and international literature [3,6,8], Gram-negative microorganisms were more frequently identified than gram-positive microorganisms (86% vs. 30%). 41% of patients had an infection sustained by Gram-negative MDR bacteria (12), *Klebsiella* spp., *Pseudomonas* spp. and *Acinetobacter* spp. the most represented (35%, 26% and 14% of microbiological isolates, respectively). The high proportion of carbapenem-resistant organisms (21% of the total isolates) confirmed the increasing trend already emerged from the ECDC and EARS-Net data relating to Italy [19,27,31]. Infections due to Gram-negative pathogens, and especially to MDR bacteria, are more frequent considering hospital-associated and ICU-associated infections. In fact, Gram-negative bacteria were isolated in 63%, 74% and 92% of patients with culture-positive infection acquired in community, hospital and ICU, respectively. Gram-negative MDR bacteria were responsible for infection in 12%, 43% and 46% of patients with culture-positive infection acquired in community, hospital and ICU, respectively (Table 2).

Probably due to the limited sample size, no microorganism was identified as independently and significantly associated with higher mortality risk. This also applies to carbapenem-resistant *Klebsiella* and *Acinetobacter* species, which are listed among the most critical antibiotic-resistant pathogens by the World Health Organization and to which a particular role in increasing the risk of death is universally attributed [31,32].

In line with the EPIC III study, even in our cohort, we found that 72% of patients received at least one systemic antimicrobial agent for prophylactic or therapeutic purposes (34.5% and 37.5% of total patients, respectively). In a significant percentage of cases, combination choices were made for both prophylaxis (32%) and therapy (72%). These data reflect an increasingly widespread but dangerous practice which, instead, deserves close monitoring, due to the high risk of developing resistance, particularly in the context of critically ill patients [33,34,35]. Given the rarity of cases in which the combined use of antibiotics allows a synergistic effect of antibiotics, the use of combined therapy with the aim of increasing the spectrum of action should be reserved for specific cases, such as multidrug-resistant pathogens treatment, to be closely monitored for prompt de-escalation [36,37,38].

Equally worthy of particular attention is the frequent use of beta-lactams in combination for prophylactic (37% of cases) instead of the therapeutic purpose of carbapenems (30% of cases) and quinolones (24%) for therapeutic purposed. Both of these practices should be carefully monitored given the ECDC, which seem to suggest, in Europe and in particular in Italy, the presence of a high resistance rate [19,27].

A final aspect of our analysis of particular interest is that our data refer to 8 different ICUs, admitting, with different modalities (emergency/scheduled), patients with different characteristics and severity (Appendix A). This obviously reflects the 60-day mortality rate, ICU and hospital LoS, and infections, since different case mixes and risk factors have a different impact on the clinical course of patients and the approach to antibiotic therapy applied by clinicians.

For this reason, on one hand, it is essential to repeat the comparison over time of the data obtained in every single ICU, taking into account the patient selection bias. On the other hand, since the ICUs included in the study are located in a similar context (i.e., the same hospital) characterized by methods for the diagnosis of infection, microbiology ecology and similar infection control and antimicrobial stewardship policies, the repeated serial comparison allows effective monitoring of the effectiveness of the corrective measured implemented over time.

Those two aspects—center-specific peculiarities on one side, homogeneity of microbiology ecology and local policies on the other side—should be considered together when planning and interpreting the results of present and future surveillance programs or interventions.

We believe that the proposal to repeatedly apply a prevalence survey tool may be particularly effective in allowing repeated comparison over time in the same (or at least similar) setting, in order to identify the emergence of new criticalities or to effectively monitor the introduction of possible corrective measures.

### Limitations

This study has limitations. First, our study collected data from eight different ICUs in a large acute-care hospital. Results are, hence, not generalizable to smaller hospitals, since infection prevalence may vary greatly with hospital beds number and case mix.

Second, to preserve the easy-to-use format of the EPIC III model [9], some aspects of infections were not approached such as timing and differentiation between acute and resolution phases. Furthermore, no data on colonization and general ICU approach to surveillance cultures were collected. Finally, no follow-up data were collected with the exception of 60 days mortality.

## 5. Conclusions

In this study, we highlighted a relatively high prevalence of infections and antimicrobial use and brought out specific critical issues relating to the different specialist ICU contexts. Considering that these aspects require continuous reassessment over time to evaluate the effects of all corrective actions implemented, we believe the repeated punctual prevalence survey represents a quick, easily repeatable, and economical method to accomplish infections and antimicrobial use surveillance in ICUs, pointing out the priorities that need improvement actions and providing feedback to health care professionals. The use of the ECDC definitions and the EPIC III protocol, known and used all over the world, is strategic to allow comparisons with national and international contexts. In addition, this surveillance might be easily repeated in the same facility, allowing monitoring of local microbiological ecology and antimicrobial use during the time to promptly identify main problematic factors and plan for specific improvement actions.

Further studies are needed to better clarify the role of prevalence investigations in infectious surveillance and their role in antimicrobial stewardship and to identify the most effective interventions to optimize antimicrobial management, especially in intensive care.

## Figures and Tables

**Figure 1 jcm-11-02482-f001:**
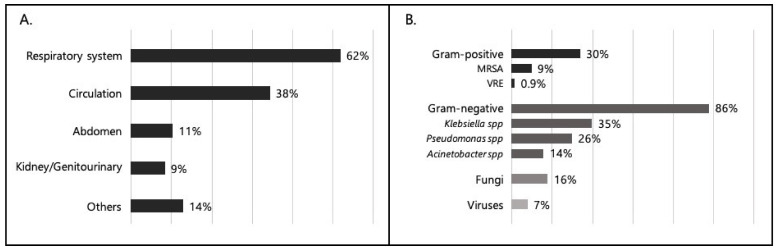
Site of infection and isolated microorganism. (**A**). Infection sites in infected patients (*N* = 184). (**B**). Isolated microorganisms in culture-positive patients (*N* = 114). Percentages can exceed 100% because patients could have more than one infection.

**Table 1 jcm-11-02482-t001:** Characteristics of patients according to the presence of infection.

		All Patients (*n* = 435)	Infection		*p* Value
			No (*n* = 251)	Yes (*n* = 184)	
Age, year, mean (SD)		57.5 (20.6)	58 (21.2)	57 (19.7)	0.4069
Male, *n* (%)		261 (60.0)	141 (56.2)	120 (65.2)	0.0572
ICU, *n* (%)	General	186 (42.8)	108 (43.0)	78 (42.4)	0.9465
	Specialist	219 (50.3)	125 (49.8)	94 (51.1)	
	Pediatric	30 (6.9)	18 (7.2)	12 (6.5)	
Type of admission, *n* (%)	Medical	120 (27.6)	46 (18.3)	74 (40.2)	<0.001 *
	Elective surgery	158 (36.3)	114 (45.4)	44 (23.9)	
	Emergency surgery	105 (24.1)	59 (23.5)	46 (25.0)	
	Trauma	52 (12.0)	32 (12.7)	20 (10.9)	
Reason for admission, *n* (%)	Respiratory	57 (13.1)	13 (5.2)	44 (23.9)	<0.001
	Cardiovascular	55 (12.6)	19 (7.6)	36 (19.6)	
	Neurological	80 (18.4)	46 (18.3)	34 (18.5)	
	Trauma	57 (13.1)	32 (12.7)	25 (13.6)	
	Surveillance	154 (35.4)	125 (49.8)	29 (15.8)	
	Other	32 (7.4)	16 (6.4)	16 (8.7)	
Comorbidities, yes, *n* (%)		274 (63.0)	159 (63.3)	115 (62.5)	0.8566
Comorbidities, *n* (%)	Solid cancer	100 (23.0)	68 (15.6)	32 (7.4)	0.0175 *
	Hematologic cancer	7 (1.6)	1 (0.4)	6 (3.3)	0.0452 *
	Diabetes Mellitus	64 (14.7)	42 (16.7)	22 (12.0)	0.1647
	COPD	54 12.4)	21 (8.4)	33 (17.9)	0.0028 *
	Heart Failure, NYHA III/IV	66 (15.2)	36 (14.6)	30 (16.3)	0.5731
	Previous cardiac disease	73 (16.8)	40 (15.9)	33 (17.9)	0.5816
	Chronic kidney failure	55 (12.6)	29 (11.6)	26 (14.1)	0.4244
	Immunosuppression	38 (8.7)	25 (10.0)	13 (7.1)	0.2908
	Solid organ transplant	39 (9.0)	26 (10.4)	13 (7.1)	0.2349
SOFA, mean (SD) ^a^		5.5 (4.1)	4.0 (3.2)	7.4 (4.5)	<0.001 *
Invasive ventilation, *n* (%)		217 (49.9)	96 (38.2)	121 (65.8)	<0.001*
Non-invasive ventilation, *n* (%)		35 (8.0)	21 (8.5)	14 (7.7)	0.7741
Tracheostomy, *n* (%)		114 (26.2)	51 (20.6)	63 (34.2)	0.0014 *
Vasopressor use, yes, *n* (%)		114 (26.2)	50 (19.9)	64 (34.8)	<0.001*
CVC, *n* (%)		372 (85.5)	206 (83.4)	166 (90.7)	0.0171 *
Urinary catheter, *n* (%)		407 (93.6)	234 (94.7)	173 (95.1)	0.7386
Renal replacement therapy, *n* (%)		39 (9.0)	11 (4.4)	28 (15.2)	<0.001 *
ECMO, *n* (%)		8 (1.8)	0 (0)	8 (4.4)	<0.001 *
Septic shock, *n* (%)		69 (15.9)	24 (9.6)	45 (24.5)	<0.001 *
Hyperlactacidemia, *n* (%)		77 (17.7)	37 (14.7)	40 (21.7)	0.0589
Antibiotic prophylaxis, *n* (%)		149 (34.3)	147 (58.6)	2 (1.1)	<0.001 *
Gastrointestinal decontamination, *n* (%) ^a^		25 (5.9)	12 (4.9)	13 (7.3)	0.2934
Chlorhexidine, *n* (%) ^a^		175 (41.6)	104 (42.4)	71 (40.3)	0.6651
ICU length of stay, days, median (IQR) ^a^		14 (4–36)	4 (1–12)	20 (10–33)	<0.001 *
Hospital length of stay, median (IQR) ^a^		29 (15–54)	11 (6–21)	50 (22–65)	<0.001 *
Mortality at 60 days, *n* (%) ^a^		52 (12.9)	19 (7.9)	33 (20.1)	<0.001 *

COPD: Chronic obstructive pulmonary disease; CVC: central venous catheter; ECMO: Extracorporeal membrane oxygenation; ICU: Intensive care unit; IQR: interquartile range; NYHA: New York Heart Association; SD: standard deviation; SOFA: Sequential Organ Failure Assessment score, * Significant at 5% level; ^a^ Total patients are not 435 because of missing values, Percentages are calculated considering missing values.

**Table 2 jcm-11-02482-t002:** Infection characteristics according to mode of acquisition and microbiological isolates (note: 184 infected patients, 114 culture-positive patients).

			Mode of Acquisition		
		Infected Patients (*n* = 184)	Community-Acquired (*n* = 36)	Hospital-Acquired/Health Care-Associated (*n* = 41)	ICU-Acquired (*n* = 117)
Evidence of infection, *n* (%)	Certain	108 (58.7)	25 (69.4)	22 (53.7)	74 (63.2)
	Probable	44 (23.9)	10 (27.8)	8 (19.5)	30 (25.6)
	Feasible	52 (28.3)	5 (13.9)	20 (48.8)	32 (27.4)
Site of infection, *n* (%)	Respiratory system	114 (62.0)	23 (63.9)	25 (61.0)	77 (65.8)
	Abdomen	21 (11.4)	3 (8.3)	9 (22.0)	10 (8.5)
	Circulation	69 (37.5)	11 (30.6)	12 (29.3)	56 (47.9)
	Kidney/genitourinary	17 (9.2)	2 (5.6)	3 (7.3)	14 (12.0)
	Others	26 (14.1)	6 (16.7)	6 (14.6)	14 (12.0)
		**Colture-Positive Patients (*n* = 114)**	**Community-Acquired (*n* = 16)**	**Hospital-Acquired/Health Care-Associated (*n* = 23)**	**ICU-Acquired** **(*n* = 85)**
Positive isolates, *n* (%)	Gram-positive	34 (29.8)	6 (37.5)	8 (34.8)	24 (28.2)
	Gram-positive MS	19 (16.7)	5 (31.3)	5 (21.7)	12 (14.1)
	Gram-positive MDR	17 (14.9)	1 (6.3)	5 (21.7)	14 (16.5)
	Gram-negative	98 (86.0)	10 (62.5)	17 (73.9)	78 (91.8)
	Gram-negative MS	69 (60.5)	9 (56.3)	9 (39.1)	56 (65.9)
	Gram-negative MDR	47 (41.2)	2 (12.5)	10 (43.5)	39 (45.9)
	All MDR bacteria	59 (51.8)	3 (18.8)	14 (60.9)	48 (56.5)
	Fungi	19 (16.7)	5 (31.3)	3 (13.0)	14 (16.5)
	Viruses	8 (7.0)	5 (31.3)	1 (4.3)	4 (4.7)
	*Klebsiella*	40 (35.1)	3 (18.8)	10 (43.5)	30 (35.3)
	*Pseudomonas*	30 (26.3)	1 (6.3)	2 (8.7)	28 (32.9)
	*Acinetobacter*	16 (14.0)	0 (0)	2 (8.7)	15 (17.6)
	Bacteria resistant to Carbapenems	36 (31.6)	1 (6.3)	8 (34.8)	30 (35.3)

ICU: intensive care unit; MDR: multi-drug resistant; MS: multi-sensitive. Percentages can exceed 100% because patients could have more than one infection.

**Table 3 jcm-11-02482-t003:** Infection characteristics according to mortality (note: total number of patients is 403, as 32 patients’ outcome data were missing).

		All Patients (*n* = 403)	Mortality at 60 Days		*p* Value
			Alive (*n* = 351)	Dead (*n* = 52)	
Antibiotic prophylaxis, *n* (%)		141 (35.2)	131 (37.4)	10 (19.6)	0.0107 *
Positive isolates, *n* (%)	Gram-positive	33 (8.2)	30 (8.5)	3 (5.8)	0.7852
	Gram-positive MS	19 (4.7)	18 (5.1)	1 (1.9)	0.4891
	Gram-positive MDR	16 (4.0)	14 (4.0)	2 (3.8)	1.0000
	Gram-negative	87 (21.6)	76 (21.7)	11 (21.2)	0.9350
	Gram-negative MS	60 (14.9)	52 (14.8)	8 (15.4)	0.9142
	Gram-negative MDR	42 (10.4)	37 (10.5)	5 (9.6)	0.8384
	All MDR bacteria	54 (13.4)	47 (13.4)	7 (13.5)	0.9888
	Fungi	16 (4.0)	9 (2.6)	7 (13.5)	0.0018 *
	Viruses	6 (1.5)	5 (1.4)	1 (1.9)	0.5659
	*Klebsiella*	36 (8.9)	35 (10.0)	1 (1.9)	0.0664
	*Pseudomonas*	28 (6.9)	25 (7.1)	3 (5.8)	1.0000
	*Acinetobacter*	15 (3.7)	13 (3.7)	2 (3.8)	1.0000
	Bacteria resistant to Carbapenems	32 (7.9)	28 (8.0)	4 (7.7)	1.0000
Site of infection, *n* (%)	Respiratory system	102 (25.3)	80 (22.8)	22 (42.3)	0.0025 *
	Abdomen	16 (4.0)	13 (3.7)	3 (5.8)	0.4460
	Circulation	56 (13.9)	40 (11.4)	16 (30.8)	0.0002 *
	Kidney/genitourinary	16 (4.0)	13 (3.7)	3 (5.8)	0.4460
	Others	24 (6.0)	21 (6.0)	3 (5.8)	1.0000
Acquisition mode, *n* (%)	Community-acquired	30 (7.4)	20 (5.7)	10 (19.2)	0.0022 *
	Hospital-acquired /Health Care-associated	36 (8.9)	27 (7.7)	9 (17.3)	0.0344 *
	ICU-acquired	106 (26.3)	87 (24.8)	19 (36.5)	0.0724

MDR: multi-drug resistant; MS: multi-sensitive; ICU: intensive care unit. * Significant at 5% level.

**Table 4 jcm-11-02482-t004:** (**A**) Univariate and multivariable logistic regression analysis with infection as the dependent variable. (**B**) Univariate and multivariable logistic regression analysis with mortality at 60 days as the dependent variable.

(A)			
		Univariate	Multivariate
Variable		OR (95% CI)	OR (95% CI)
Age		1.00 (0.99–1.01)	1.00 (0.99–1.01)
Gender	Male	1 (reference)	1 (reference)
	Female	0.69 (0.47–1.02)	0.92 (0.57–1.48))
Reason for admission	Respiratory	1 (reference)	1 (reference)
	Cardiovascular	0.56 (0.24–1.29)	0.37 (0.15–0.92)
	Neurological	0.22 (0.10–0.47)	0.23 (0.10–0.53)
	Trauma	0.23 (0.10–0.52)	0.29 (0.12–0.68)
	Surveillance	0.07 (0.03–0.15)	0.07 (0.03–0.15)
	Other	0.30 (0.12–0.75)	0.25 (0.09–0.69)
Invasive ventilation		3.13 (2.11–4.66)	2.14 (1.29–3.53)
Vasopressor use		2.19 (1.42–3.38)	1.53 (0.86–2.73)
CVC		1.97 (1.10–3.54)	1.37 (0.68–2.74)
Renal replacement therapy		3.90 (1.89–8.06)	2.78 (1.30–5.96)
COPD		2.38 (1.33–4.28)	2.15 (1.06–4.35)
**(B)**			
**Variable**		**OR (95% CI)**	**OR (95% CI)**
Age		1.04 (1.02–1.07)	1.05 (1.02–1.08)
Gender	Male	1 (reference)	1 (reference)
	Female	1.20 (0.66–2.16)	2.20 (1.07–4.53)
Source of admission	Operating room/Surgical department	1 (reference)	1 (reference)
	Emergency department	1.13 (0.51–2.48)	1.03 (0.38–2.78)
	Medical department	3.56 (1.53–8.26)	3.69 (1.39–9.78)
	Other hospital	0.86 (0.27–2.69)	0.59 (0.15–2.35)
	Other ICUs	2.01 (0.76–5.29)	3.86 (1.16–12.84)
Comorbidities		11.53 (3.52–37.73)	12.77 (2.91–56.02)
Invasive ventilation		4.71 (2.34–9.48)	4.22 (1.91–9.30)
Site of infection	Circulation	3.56 (1.81–7.00)	3.43 (1.48–7.97)
Acquisition mode	Community-acquired	4.04 (1.77–9.22)	9.90 (3.07–31.92)

CI: confidence interval; COPD: Chronic obstructive pulmonary disease; CVC: central venous catheter; OR: odds ratio. ICU: intensive care unit.

**Table 5 jcm-11-02482-t005:** Characteristics of patients according to mortality (note: total number of patients is 403, as in 32 patients outcome data were missing).

		All Patients (*n* = 403)	Mortality at 60 Days		*p* Value
			Alive (*n* = 351)	Dead (*n* = 52)	
Age, year, mean (SD)		57.5 (20.6)	56.5 (22.9)	66.6 (15.0)	<0.001 *
Male, *n* (%)		236 (58.6)	208 (59.3)	28 (53.8)	0.4596
ICU, *n* (%)	General	173 (42.9)	146 (41.6)	27 (51.9)	<0.001 *
	Specialist	203 (50.4)	180 (51.3)	23 (44.2)	
	Pediatric	27 (6.7)	25 (7.1)	11 (3.8)	
Type of admission, *n* (%)	Medical	109 (27.0)	87 (24.8)	22 (42.3)	0.0122 *
	Surgical election	151 (37.5)	135 (38.5)	16 (30.8)	
	Surgical emergency	95 (23.6)	82 (23.4)	13 (25.0)	
	Trauma	48 (11.9)	47 (13.4)	1 (21.9)	
Reason for admission, *n* (%)	Respiratory	52 (13.0)	41 (11.7)	11 (21.2)	<0.001 *
	Cardiovascular	50 (12.4)	36 (10.3)	14 (26.9)	
	Neurological	72 (17.9)	64 (18.2)	8 (15.4)	
	Trauma	53 (13.2)	52 (14.8)	1 (1.9)	
	Surveillance	147 (36.2)	134 (37.8)	13 (25.0)	
	Other	29 (7.2)	24 (6.8)	5 (17.2)	
Comorbidities, yes, *n* (%)		252 (62.5)	204 (58.1)	48 (92.3)	<0.001 *
Comorbidities, *n* (%)	Solid cancer	94 (23.3)	77 (19.1)	17 (4.2)	0.0870
	Hematologic cancer	6 (1.5)	3 (0.9)	3 (5.8)	0.0306 *
	Diabetes Mellitus	60 (14.9)	50 (14.2)	10 (19.2)	0.3459
	COPD	50 (12.4)	35 (10.0)	15 (28.8)	<0.001 *
	Heart Failure, NYHA III/IV	59 (14.6)	46 (13.1)	13 (25.0)	0.0236 *
	Previous cardiac disease	68 (16.9)	47 (13.4)	21 (40.4)	<0.001 *
	Chronic kidney failure	48 (11.9)	36 (10.3)	12 (23.1)	0.0077 *
	Immunosuppression	36 (8.9)	30 (8.5)	6 (11.5)	0.4405
	Solid-organ transplant	36 (8.9)	31 (8.8)	5 (9.6)	0.7965
SOFA, mean (SD) ^a^		5.5 (4.1)	4.5 (3.6)	9.6 (4.1)	<0.001*
Invasive ventilation, *n* (%)		194 (48.1)	153 (43.6)	41 (78.8)	<0.001 *
Non-invasive ventilation, *n* (%)		33 (8.3)	28 (8.1)	5 (9.6)	0.5971
Tracheostomy, *n* (%)		101 (25.3)	87 (25.0)	14 (26.9)	0.7401
Vasopressor use, yes, *n* (%)		101 (25.1)	75 (21.4)	26 (50.0)	<0.001 *
CVC, *n* (%)		343 (86.0)	294 (84.7)	49 (94.2)	0.0478 *
Urinary catheter, *n* (%)		376 (94.7)	325 (94.2)	51 (98.1)	0.2300
Renal replacement therapy, *n* (%)		36 (8.9)	23 (6.6)	13 (25.0)	<0.001 *
ECMO, *n* (%)		7 (1.8)	6 (1.7)	1 (1.9)	1.0000
Septic shock, *n* (%)		58 (14.4)	42 (12.0)	16 (30.8)	<0.001 *
Hyperlactacidemia, *n* (%)		64 (15.9)	47 (13.4)	17 (32.7)	<0.001 *
ICU length of stay, days, median (IQR) ^a^		14 (4–35)	14 (3–34)	14 (8–45)	0.2494
Hospital length of stay, median (IQR) ^a^		29 (16–54)	28 (16–52)	39 (14–68)	0.4315

COPD: Chronic obstructive pulmonary disease; CVC: central venous catheter; ECMO: Extracorporeal membrane oxygenation; ICU: Intensive care unit; IQR: interquartile range; NYHA: New York Heart Association; SD: standard deviation; SOFA: Sequential Organ Failure Assessment score. * Significant at 5% level; ^a^ Total patients are not 435 because of missing values. Percentages are calculated considering missing values.

## Data Availability

Data available on reasonable request from the authors.

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
