# Peer review of "Effectiveness of an Active and Continuous Surveillance Program for Intensive Care Units Infections Based on the EPIC III (Extended Prevalence of Infection in Intensive Care) Approach"

_jcm, 2022, doi:10.3390/jcm11092482_

Round 1
Reviewer 1 Report
Thank you for the opportunity to review this interesting article entitled: “Effectiveness of an active and continuous surveillance program for Intensive Care Units infections based on the EPIC III (Extended Prevalence of Infection in intensive Care) approach” by Montrucchio et al.
The authors present a single center study (Hospital of Turin) on the prevalence of infection among patients admitted into 8 local ICU (72 ICU beds). They did a 24-hour point prevalence each 2 months, for one-year study period, totalizing 6 days of data collection, utilizing the same CRF previously used in EPIC III study.
Considering that the authors report epidemiologic microbiology results from a single hospital, these have limited application outside the institution and relevant mainly to the local healthcare players.
Italy is listed as a member of the Healthcare-Associated Infections Surveillance Network (HAI-Net), coordinated by the European Centre for Disease Prevention and Control (ECDC). I do not know if Hospital of Turin is part of that network, if so it would be highly interesting to add a comparison of results collected by each method; which would increase the interest of the manuscript, particularly concerning ICU-acquired infections.
SPECIFIC COMMENTS BY SECTION
Introduction
- Page 1, 2nd paragraph: I cannot understand the relation between both phrases, please clarify.
- Page 2, 1st paragraph: does not add background for current manuscript, considering that authors do not provide data on adherence to this surveillance system, could be suppressed.
- Please simplify the objectives in the last paragraph of “Introduction”, there is repeated information.
Materials and methods
- Line 88: the authors did the follow-up up to 60 days regardless of the patient location or do they mean up to 60 days or hospital discharge? Please clarify.
- The “2 Study context” (line 94) is not relevant for current study. Could be deleted.
- “2.3 Data collection” (line 102) please provide the CRF as supplemental data or electronic link and just mentioned in the manuscript the relevant differences to the original CRF. Who collected the data and how (paper or electronic)?
- “2.4. Operative definitions” (line 111) – please use always the same term to refer to hospital-acquired and nosocomial infections (these terms are used interchangeable throughout the text and that confounds the reader). Please define what you considered antibiotic prophylaxis.
Results
- Please mentioned the number of patients excluded for absence of informed consent, to give any idea of risk of bias.
- I have great doubts about including pediatric patients in the analysis, it is a different patient population with very different epidemiology and guidelines of infection prevention and treatment. I suggest that data regarding pediatric patients be deleted.
- Well-built table 1, but authors should calculate p value for each item and not for the group if they really want to draw attention for those settings associated with more infection (like type and reason of admission).
- As you did with antibiotic prophylaxis please add GI decontamination, chlorhexidine and nasal mupirocin to table 1.
- Table 1A. should be moved to supplemental data. Except for internal hospital use I do not see the interest in the individual data of the units.
- Line 174: do you mean hospital-acquired (as nosocomial) infection or healthcare associated infection as defined by Friedman et al: “Health care--associated bloodstream infections in adults: a reason to change the accepted definition of community-acquired infections” in Ann Intern Med. 2002 Nov 19;137(10):791-7.? Please maintain the same technical name along the manuscript.
- Please add another table with data in columns regarding place of acquisition/acquisition mode (CA, HA or ICU-acquired) and for each category (in lines) provide the evidence of infection, site or focus of infection, isolation rate and microbiological data (similar to table 2). It is very difficult to interpret microbiological data when not taking in consideration focus of infection and place of acquisition. By adding the new table, please simplify results in page 8, lines 178-186 and subtract figure 1.
- There are inconsistencies between tables and results description: in table 1A the authors consider a total of 435 patients and 158 isolates (34 gram+, 97 gram-, 19 fungi and 8 virus); in table 2, a total of 403 patients and 142 isolates (33 gram+, 87 gram-, 16 fungi and 6 virus). In lines 193-194 the authors state: “overall number of isolates (total =170)”. Please clarify these discrepancies and rectify the rate of MDR and resistant to carbapenems microorganisms.
- Please add in the text a reference to the MRSA isolations.
- When referring to resistant to carbapenems are authors referring to carbapenemase producing enterobacteraceae? These are bacteria of interest, if you are in possession of such data please report, along with ESBL producing bacteria, and MDR Pseudomonas and Acinectobacter.
- “2. Antibiotic therapy” (line 206) – please report the use of antibiotic therapy according to type of infection: community, hospital or ICU-acquired.
- Do not see the advantage of study the association of general risk factors with “general” infection under the context of this manuscript. If the authors really want to proceed please do it separately for community, hospital and ICU-acquired infection since they are associated with completely different risk factors.
- Regarding mortality analysis, I suggest the authors complete table 4, adding a column with crude OR for each variable, and then develop the multivariable logistic regression with the ones considered statistically associated with the outcome and those clinically relevant (like the presence or absence of infection) and present just the variables retained in the final model, with the adjusted OR and the 95%CI.
Discussion and conclusions
- Should be reviewed considering what has been exposed.
Reviewer 2 Report
In the paper “Effectiveness of an active and continuous surveillance program for Intensive Care Units infections based on the EPIC III (Extended Prevalence of infection in Intensive Care) approach” the authors have evaluated the effectiveness of the EPIC III data collection protocol as an active surveillance tool in ICUs. They collected the data from eight ICUs of the University Hospital of Turin and included a total of 435 patients. Their data confirm a high prevalence of infections, sepsis and use of antimicrobials in ICU settings. They have concluded that repeated punctual prevalence survey seems an effective method to carry out the surveillance of infections and the use of antimicrobials in the ICU. The use of the EPIC III protocol is also allowing comparisons between the ICUs at the national and international level.
Their study offers valuable data for mortality, length of ICU and hospital stay and evaluation of EPIC III protocol. The presented data prompt for more studies to identify the most effective interventions to optimize antimicrobial management, especially in intensive care settings. However, data presentation in the manuscript should be improved for clarity and in interest of audience.
Comments
- Table 1. Should be reformatted allowing for clear visibility of columns with patient data.
- Table 1A is also hard to follow. Should be re-formatted or divided into 2 or more Tables.
- Figure 1 A&B should be bigger as well as the size of letters and numbers in the Figures.
- Most important data from the Tables should be presented in Figure format for clarity; will be easier for audience to follow.
Round 2
Reviewer 2 Report
The authors considerably improved the manuscript and clarity of presented Tables and Figures. It's much easier for audience to follow this otherwise interesting and important work.